# Study on Reversible Solubilization by Adjusting Surfactant Properties

**DOI:** 10.3390/ma16093550

**Published:** 2023-05-05

**Authors:** Youichi Takata, Amu Uchikura

**Affiliations:** Department of Chemical and Biological Engineering, National Institute of Technology, Ube College, Yamaguchi 755-8555, Japan

**Keywords:** amphoteric surfactant, surface tension, micelle, reversible solubilization, controlled release

## Abstract

Solubilization allows us to dissolve hydrophobic materials in water and to carry them to where they are needed. The purpose of this study is to control solubilization, especially the release of solubilized materials, via external stimulation. An amphoteric surfactant, dodecyldimethyl(3-sulfopropyl)ammonium hydroxide inner salt (SB-12), was employed, and a pH change was chosen as the external stimulus. We measured the surface tension of an SB-12 solution via the Wilhelmy method, and the absorbance of a solubilized solution was determined using UV-Vis spectroscopy at various pH values. The surface tension was almost the same at any pH, contrary to our expectations. This result suggests that the adsorption behavior and micelle formation of SB-12 were not affected by pH very much. On the other hand, the solubilization behavior remarkably depended on the pH. In particular, the solubilization ability under the basic condition was much larger than that under the acidic and neutral conditions. Taking advantage of such a difference in solubilization ability under some pH conditions, the solubilized material could be completely removed from the solution. Thus, we clarified the mechanism of release for solubilized materials due to a pH change.

## 1. Introduction

Surfactants that are widely used as detergents have both hydrophilic and hydrophobic groups in their molecular structure. When surfactants are dissolved in water, surfactant molecules gather and form aggregates. The inside of these aggregates consists of hydrophobic groups to avoid contact with water. On the other hand, the outside of these aggregates is covered by hydrophilic groups. Such an aggregate is called a micelle. Additionally, the concentration at which micelle formation is initiated in a solution is referred to as the critical micelle concentration (CMC).

The inside of micelles resembles a microscopic pool of hydrophobic materials. Therefore, micelles in aqueous solutions can encapsulate hydrophobic materials that are inherently insoluble in water. This phenomenon is called solubilization and is the most fundamental function in colloid and interface science. Solubilization has been utilized in many industrial fields such as pharmaceutics, cosmetics, foods, pigments, fuels, etc. In these fields, hydrophilic materials such as water are often in contact with hydrophobic materials such as oil. When the incorporation of hydrophobic materials into hydrophilic materials or vice versa is required, solubilization is very useful.

A recent study analyzed solubilization from the standpoint of sustained release. Sustained release means that solubilized materials (solubilizates) are gradually released from micelles. For example, sustained release is a drug carrying mechanism in pharmaceutics. Many drugs have hydrophobic properties, and it is not easy to dissolve them in water. In order to improve these drugs’ solubility into water, solubilization has been studied. In addition, sustained release is a useful function of solubilization. It is expected that the effects of a drug will continue for a long time if we can control the sustained release of solubilizates via solubilization. Researchers have attempted to control sustained release based on various points of view: by mixing polymers with surfactants [1,2], adjusting the molecular weight of the polymers [3], adding salts [4], combining silicone surfactant micelles with agar hydrogels [5], etc.

In the studies described above, the solubilizates were spontaneously released from the micelles over time, but there were some problems with the release behavior. The incorporation of hydrophobic materials led to the stabilization of micelles, and the release effect was often weakened [6,7]. To address this problem, other researchers have been attempting to control the release of solubilizates more actively. However, those methods usually require external stimuli, which thus requires analyzing whether external stimuli affect the stability of micelles. There are various candidates that can act as external stimuli [8,9,10,11]: temperature, electrical potential, light, pH, magnetic field, CO_2_ concentration, etc. For example, surfactants, including azobenzene, change their properties via irradiation with ultraviolet and visible light [12,13,14]. This is because light irradiation results in a transformation between the cis- and trans-forms of azobenzene. Such photoresponsive surfactants allow us to regulate micelle formation due to light irradiation.

Thus, it is profitable for future applications to be able to control micelle formation instantaneously and reversibly. In this study, we aim to clarify the factors affecting the stability of micelles, to finely control solubilization using those factors, and to establish an instant and reversible mechanism of release and uptake of insoluble materials. Controlling solubilization can be useful in the field of oil recovery [15], as well as in pharmaceutics. Additionally, the establishment of a reversible mechanism can be linked to cyclic solubilization [16].

There are two methods to control solubilization. One is to take advantage of the difference in solubilization power under different environments, and the other is to destroy the micelle itself due to external stimuli. In our previous study on the controlled release of solubilizates, we focused on the former method, i.e., the difference in the hydrophobicity between hydrocarbon and fluorocarbon materials [17]. It is well known that hydrocarbon materials are hardly miscible with fluorocarbon materials. Thus, such a low miscibility brings about immiscibility in the mixed micelle of hydrocarbon and fluorocarbon surfactant mixture systems [18,19]. Therefore, we can control the formation of hydrocarbon-rich and fluorocarbon-rich micelles by adjusting the composition of hydrocarbon and fluorocarbon surfactants in an aqueous solution. It was also expected that the hydrocarbon-rich micelles would solubilize the hydrocarbon materials, whereas the fluorocarbon-rich micelles were expected to not. However, the hydrocarbon surfactant lithium dodecyl sulfate (LiDS) solubilized the hydrocarbon material Sudan III, but the fluorocarbon surfactant heptadecafluoro-1-octanesulfonate (LiFOS) did not solubilize Sudan III at all. This result suggests the possibility of controlled release of solubilizates, utilizing the difference in solubilization behavior caused by hydrophobicity. Thus, we prepared a LiDS aqueous solution, which solubilized Sudan III and added the LiFOS aqueous solution to it. As a result, the solubilized Sudan III was released from the micelles of the LiDS associated with the addition of the LiFOS aqueous solution. Therefore, the difference in hydrophobicity of the surfactants allowed us to release the solubilizates. On the other hand, the release behavior was not sensitive to the addition of the LiFOS aqueous solution, and it took some time to release Sudan III, although we expected a drastic change.

The above results were significant for the sustained release of solubilizates. From the standpoint of controlled release, we also required a mechanism of instantaneous release as a kind of switch. Then, we focused on the second method of controlled solubilization, i.e., the effect of pH on the release of solubilizates. This was because it was easy for us to adjust the pH in a solution and to alter the solution’s condition quickly and extensively and because a pH-responsive effect had, in fact, been previously employed in the field of drug delivery and controlled release [20,21]. If the quantity of solubilizates depends on pH, a change in pH is more likely to induce the release of solubilizates. In this respect, we chose an amphoteric surfactant that included both cations and anions in its hydrophilic group. In a similar manner to amino acids, amphoteric surfactants can be cationic or anionic surfactants depending on pH [22]. Via such a change in the properties of amphoteric surfactants, we tried to establish the controlled release of the solubilizates.

The aim of this study was to determine the controlled release mechanism of solubilizates. We used amphoteric surfactants, the properties of which depended on pH. Namely, the micelle formation of amphoteric surfactants depended on pH. Taking advantage of this property, we could control solubilization using pH.

## 2. Materials and Methods

### 2.1. Materials

The amphoteric surfactant dodecyldimethyl(3-sulfopropyl)ammonium hydroxide inner salt (sulfobetaine-12, SB-12) was purchased from Tokyo Chemical Industry Co., Ltd. (Tokyo, Japan) and used without further purification. The reddish-brown dye Sudan III (FUJIFILM Wako Pure Chemical Corporation, Osaka, Japan) was used as a solubilizate. The solution at various pH values was prepared using distilled water, a 0.1 mol L^−1^ hydrochloric acid solution (FUJIFILM Wako Pure Chemical Corporation, Osaka, Japan), and a 0.1 mol L^−1^ sodium hydroxide solution (FUJIFILM Wako Pure Chemical Corporation, Osaka, Japan).

### 2.2. Surface Tension Measurement

The surface tension measurements were performed using an automatic surface tensiometer DY-300 (Kyowa Interface Science Co., Ltd., Saitama, Japan) based on the Wilhelmy plate method. The SB-12 solution was prepared with distilled water, a 0.1 mol L^−1^ hydrochloric acid solution, and a 0.1 mol L^−1^ sodium hydroxide solution. The surface tension was measured as a function of the molarity of SB-12.

### 2.3. pH Measurement

The values of pH at each solution were measured using a pH meter AS800 (AS ONE Corporation, Osaka, Japan). The pH of the solution containing solubilized Sudan III was not measured to avoid contamination of the electrodes using the pH meter. We assumed that the effect of Sudan III on pH was negligibly small because of its quite low solubility in an aqueous solution.

### 2.4. Solubilization Measurement

To measure the absorbance of SB-12 solutions containing solubilized Sudan III, a high-precision UV-VIS spectrophotometer PD-3000UV (APEL Co., Ltd., Saitama, Japan) was used. The wavelength was fixed at 520 nm. The temperature throughout the experiments was kept at 298.15 K using a water bath BK300 (Yamato Scientific Co., Ltd., Tokyo, Japan). The absorbance of the SB-12 solutions containing solubilized Sudan III was measured as functions of the molarity of SB-12 and pH.

The samples were prepared according to the following procedures: (1) SB-12 solutions of each concentration were prepared at a fixed pH. (2) Sudan III was added to the prepared sample, and the solution was well stirred with a vortex mixer VTX-3000L (AS ONE Corporation). (3) The sample was kept in a water bath at 298.15 K for 30 min. (4) The sample was filtered to remove any excess Sudan III. (5) The absorbance of the filtrate was measured at 520 nm.

### 2.5. Controlled Release Measurement

The release of solubilized Sudan III from the micelles was performed according to the following procedure: (1) A sample of the 5 mmol L^−1^ SB-12 solution was prepared at pH = 12.5. (2) Sudan III was added to the prepared sample, and the solution was well stirred with a vortex mixer. (3) The sample was kept in a water bath at 298.15 K for 30 min. (4) The sample was filtered to remove excess Sudan III. (5) Distilled water, a hydrochloric acid solution, or a sodium hydroxide solution was added to the filtrate. Then, the sample was well stirred with a vortex mixer and kept in a water bath at 298.15 K for 30 min. (6) The sample was filtered to remove Sudan III that had been deposited from the solution after the addition of distilled water, a hydrochloric acid solution, or a sodium hydroxide solution. (7) The absorbance of the filtrate was measured at 520 nm.

## 3. Results and Discussion

### 3.1. Surface Tension of SB-12 Solution

The surface tension measurements of the surfactant solutions elucidated the adsorption behavior and the micelle formation of surfactants. To clarify the properties of SB-12, we measured the surface tension of the SB-12 solutions.

For the surface tension measurement, distilled water, a 0.1 mol L^−1^ hydrochloric acid solution, and a 0.1 mol L^−1^ sodium hydroxide solution was used as the solvent. This enabled the evaluation of the pH effect on the properties of SB-12. Prior to the surface tension measurement, we measured the pH of each sample. The results were 1.2, 6.0, and 12.5 for distilled water, the 0.1 mol L^−1^ hydrochloric acid solution, and the 0.1 mol L^−1^ sodium hydroxide solution, respectively. It was also confirmed that the variation in pH was negligibly small within the concentration of SB-12 prepared in this study. Therefore, in this study, we assumed that pH was constant, irrespective of the concentration of SB-12, so long as the solvent was a common solution.

Figure 1 shows the dependence of the surface tension on the molarity of SB-12 in distilled water (pH = 6.0). It was found that the surface tension steeply decreased with increasing molarity in the lower concentration region and became constant above 3.35 mmol L^−1^. Therefore, the CMC value of SB-12 in distilled water was 3.35 mmol L^−1^.

Next, we estimated the surface density *Γ* of SB-12 at the air/water surface to see the adsorption behavior. The surface density showed the excess quantity of surfactants at the air/water surface compared with their quantity in the aqueous solution. It could be calculated by applying the following equation to the surface tension *γ* versus the molarity *C* curves at the constant temperature *T* and pressure *p* [23]:(1)$$\Gamma =-\frac{C}{RT}{\left(\frac{\partial \gamma }{\partial C}\right)}_{T,p},$$
where *R* represents the gas constant. Figure 2 shows the surface density versus molarity of SB-12 in distilled water (pH = 6.0) curve. The surface density of surfactants at the air/water surface generally increases due to the adsorption of surfactants on the surface. Therefore, the surface density increases with increasing molarity of SB-12. In particular, the steep increase in surface density at the lower concentration region in Figure 2 suggests a strong adsorption ability of SB-12 at pH = 6.0. Then, the surface density gently increases and approaches about 2.5 μmol m^−2^.

To clarify the state of the adsorbed film, furthermore, we calculated the surface pressure *Π* and the surface area *A* occupied by an adsorbed SB-12 molecule using the following equations:(2)$$\Pi =\gamma^{0}-\gamma ,$$
(3)$$A=\frac{1}{N_{A}\Gamma },$$
where $\gamma^{0}$ and *N*_A_ stand for the surface tension of the air/pure water surface and Avogadro’s number, respectively. The *Π* versus *A* curve is depicted in Figure 3. It was found that the *Π* values increased with decreasing *A* values. The *Π* versus *A* curve in two-dimensions corresponded to the pressure versus molar volume curve in a three-dimensional space. In other words, the adsorbed film was mainly classified into three kinds of states analogous with the three-dimensional states: the gaseous, expanded, and condensed states. The adsorbed film of SB-12 at pH = 6.0 was judged to be the expanded state, referenced from a previous study [24,25]. It is also worth noting that the minimum value of *A* was about 0.68 nm^2^. The cross section of a surfactant molecule with one hydrophobic chain was about 0.20~0.30 nm^2^. It was suggested from this result that the adsorbed film of SB-12 at pH = 6.0 was packed loosely to a certain extent, even at a higher concentration region. The dipole moment occurs between the cation and anion in the hydrophilic group of SB-12 [26]. The repulsive interaction due to such a dipole moment acted between the adsorbed molecules at the air/water surface. Therefore, the minimum value of *A* in this system became larger than the value expected for close packing.

Next, we measured the surface tension of the SB-12 solution at pH = 1.2 (0.1 mol L^−1^ hydrochloric acid) and 12.5 (0.1 mol L^−1^ sodium hydroxide). The results are shown in Figure 4. The concentration dependence of surface tension at pH = 1.2 was almost the same as that at pH = 12.5. Both results were slightly lower compared with the result at pH = 6.0. The CMC values of SB-12 in solution were 2.50 mmol L^−1^ at pH = 1.2 and 2.01 mmol L^−1^ at pH = 12.5.

Furthermore, we estimated the surface density, the surface pressure, and the surface area by applying Equations (1)–(3) to Figure 4. Figure 5a shows the surface density versus molarity curves, and Figure 5b shows the surface pressure versus surface area curves at pH = 1.2, 6.0, and 12.5. From these results, we clearly found that the adsorption behaviors of SB-12 at pH = 1.2 and 12.5 were similar to each other. Additionally, the maximum values of surface density at pH = 1.2 and 12.5 were a little lower than that at pH = 6.0. In other words, the minimum values of surface area at pH = 1.2 and 12.5 were a little larger than those at pH = 6.0. This implied that SB-12 molecules were loosely packed at the air/solution surface at pH = 1.2 and 12.5, compared with pH = 6.0. The amphoteric surfactants had both cations and anions in their hydrophilic groups. Then, the charged state of the hydrophilic group depended on pH. For example, the hydrophilic groups were positively charged under the acidic condition but negatively charged under the basic condition, analogous to an amino acid. As stated above, the repulsive interaction due to the dipole moment in the hydrophilic group of SB-12 existed under the neutral condition, and it caused a loosely packed adsorbed film. On the other hand, electrostatic repulsion occurred between adsorbed molecules due to the charge of the hydrophilic groups under the acidic or basic conditions. This effect may have been stronger than that of the dipole moment. This was responsible for the more loosely packed adsorbed film at pH = 1.2 and 12.5 than that at pH = 6.0.

From the viewpoint of micelle formation, the CMC values at pH = 1.2, 6.0, and 12.5 were 2.50, 3.35, and 2.01 mmol L^−1^, respectively. There was a slight difference among them. However, the pH dependence of the CMC values seemed to be strange. In general, the CMC values of the ionic surfactants were larger than those of nonionic surfactants because of the repulsion force between the hydrophilic groups. Therefore, it was expected that the CMC values at pH = 1.2 and/or 12.5 would become larger than those at pH = 6.0, but we obtained the opposite results. One speculated reason was that the repulsive force between the hydrophilic groups of SB-12 due to the dipole moment occurred even under the neutral condition, as mentioned above. Another reason was that the micelle surface had a curvature. This allowed the hydrophilic groups of SB-12 to be energetically favorably oriented [27]. These factors brought about opposite results in terms of the CMC.

Judging from the results of the surface tension measurement, the properties of the micelle formation of SB-12 were scarcely dependent on pH. Next, we evaluated the solubilization behavior of the SB-12 solution at each pH value.

### 3.2. Solubilization of Sudan III into SB-12 Solution

Once the surfactant molecules formed micelles in the solution, i.e., the solution concentration exceeded the CMC values, solubilization took place. We measured the absorbance of solubilized Sudan III as functions of solution concentration and pH. As a result, it was found that the solubilization behavior remarkably depended on pH.

Figure 6a shows the absorbance of the solution at each pH against the concentration of SB-12. It was found that the absorbance linearly increased with increasing concentration of SB-12 at all pH values. This meant that Sudan III was undoubtedly solubilized in the SB-12 solution and that the solution became red. Compared with the results of the surface tension measurements, the absorbance increased to above the CMC values at each pH value. This indicated that solubilization occurred alongside micelle formation. Looking closely at the absorbance behavior, it was found that the concentration dependence of absorbance at pH = 1.2 and 6.0 was almost identical, whereas those at pH = 11.0, 12.0, 12.3, and 12.5 were remarkably different from each other. Here, the maximum value of the absorbance for this apparatus was reached in the high concentration range at pH = 11.0, 12.0, 12.3, and 12.5. From these results, it looks as if Sudan III was solubilized more under the basic condition.

Figure 6b shows the pH dependence of absorbance at a constant concentration for the SB-12 solution. When the concentration is lower than the CMC values, there is no difference in absorbance for all pH values (curve 1). On the other hand, as the concentration increases, the absorbance in the vicinity of pH = 12 steeply increases (curves 2–6). This also suggests that Sudan III can be sufficiently solubilized under the basic condition.

The larger solubilization power under the basic condition may be attributed to the structural change in Sudan III. A series of Sudan compounds can take the azo and hydrazone forms depending on the pH [28,29]. The azo form under the acidic condition has no charge, whereas the hydrazone form under the basic condition has a negative charge. Such a negative charge can interact with the positive charge in the hydrophilic group of SB-12. This effect stabilizes the solubilization of Sudan III and leads to the larger solubilization power under the basic condition. The structural change in Sudan III changes the color of the solubilized solution from red to purple, as shown in Figure 7.

The results in Figure 6 suggest that the extent of solubilization depends on pH, and, furthermore, the change from the basic to acidic conditions may release excess Sudan III.

### 3.3. Release of Solubilized Sudan III from Micellar Solutions

In the above section, we found that the SB-12 solution sufficiently solubilized Sudan III under the basic condition. Therefore, we can infer that Sudan III was removed from the solution when the solution property changed from basic to acidic conditions. Then, we conducted an experiment on the release of solubilized Sudan III from the micellar solutions due to pH changes.

Figure 7 shows the appearance of the filtrate after water or hydrochloric acid were added into the basic solution containing solubilized Sudan III. As 0.1 mol L^−1^ hydrochloric acid was added (Figure 7a), the filtrate became colorless and transparent. From this result, it was found that the solubilized Sudan III was completely removed from the solution due to the addition of 0.1 mol L^−1^ hydrochloric acid. The addition of 0.05 mol L^−1^ hydrochloric acid also enabled the filtrate to become colorless and transparent, although more hydrochloric acid was needed (Figure 7b). When we added 0.01 mol L^−1^ hydrochloric acid or water (Figure 7c,d), the color of the filtrate became light but still remained red. This suggested that the solubilized Sudan III was not completely removed from the solution.

Figure 8 shows the absorbance of the filtrate in Figure 7 against the volume ratio of added water or various concentrations of hydrochloric acid to the whole solution after mixing. Before the addition of water or hydrochloric acid, the solution sufficiently solubilized Sudan III, and the absorbance reached the limit of the apparatus. Once water or hydrochloric acid were added into the solution, the absorbance of the filtrate decreased. After the addition of 0.1 mol L^−1^ hydrochloric acid (Figure 8a), the absorbance greatly decreased to around a volume ratio of 0.5, which meant that the volume of the originally prepared solution was equal to the added volume of the 0.1 mol L^−1^ hydrochloric acid, and the pH was close to 7. Therefore, it can be presumed that the color of the solution became transparent because of the neutralization of the solution. Looking closely at Figure 8a, the absorbance behavior was certainly similar to the pH change. As shown in Figure 8b, we also obtained the same results in the case of the addition of 0.05 mol L^−1^ hydrochloric acid. However, we needed more hydrochloric acid because of its dilution. Again, the absorbance approached zero at around pH = 7. As the 0.1 mol L^−1^ sodium hydroxide solution was neutralized using 0.05 mol L^−1^ hydrochloric acid, it was found from a simple calculation that the volume ratio was 0.667. This corresponded to the results in Figure 8b. When 0.01 mol L^−1^ hydrochloric acid or water were added, on the other hand, the absorbance linearly decreased with increasing volume ratio (Figure 8c,d), but the filtrate was not sufficiently colorless, as can be seen from Figure 7c,d. It is thus hard to conclude that Sudan III was removed perfectly. In this case, the solution was only diluted. As we further added 0.05 mol L^−1^ hydrochloric acid, the color of the filtrate finally became colorless above a volume ratio of 0.900, which was comparable with that when the 0.1 mol L^−1^ sodium hydroxide solution was neutralized by 0.01 mol L^−1^ hydrochloric acid. In contrast, the further addition of water never caused the filtrate to become colorless. From these results, we found that a pH value less than 7 was required to release solubilized Sudan III.

The reason why a pH value less than 7 was needed for the release of Sudan III could be speculated as follows. At first, from Figure 7d, we found that the micelle was not broken even when the solution was sufficiently diluted once the micelle solubilized Sudan III. For example, solutions at volume ratios of 0.667 and 0.833 in Figure 7d were diluted to three-fold and six-fold, respectively. These concentrations were lower than that determined via CMC using the surface tension measurement, but Sudan III was still solubilized in the solution. This meant that micelles still existed in the solution. This phenomenon may be affected by two factors. One was the increase in cohesive force of the core of micelles due to the existence of hydrophobic Sudan III, and the other was the stabilization of micelles due to the attraction between the hydrophilic group of SB-12 and the negative charge of Sudan III, as mentioned above [28,29]. In particular, the latter factor altered the position of solubilized Sudan III in the micelle. The hydrophobic materials were usually solubilized in the core of micelles, but the deprotonation of Sudan III placed Sudan III at the palisade region in the micelle [30]. When hydrochloric acid was added to the solubilized solution under the basic condition, Sudan III protonated with the decrease in pH. The complete protonation of Sudan III remarkably lowered the solubilization power, and, as a result, Sudan III could be completely released from the solution.

## 4. Conclusions

In order to actively control solubilization via a pH change, we investigated the properties of the amphoteric surfactant SB-12 and its solubilization behavior associated with the pH change. From the surface tension measurement, it was found that the adsorption behavior depended on the pH. This was attributable to the dipole moment and the charged state in the hydrophilic group of SB-12. On the other hand, the micelle formation was similar irrespective of pH. However, the solubilization behavior remarkably depended on the pH. The solubilization power under the basic condition was fairly large compared with that under the acidic and neutral conditions. The protonation/deprotonation of Sudan III solubilized in the micelle resulted in such differences. Thus, the change in the solution from the basic to the acidic conditions allowed us to completely release the solubilized Sudan III from the solution. Specifically, a pH value less than seven was required for the release of solubilized Sudan III in this system.

It was found that we could easily and instantaneously control the solubilization behavior by using an amphoteric surfactant and adjusting the pH. Based on this study, we will try to more finely control the solubilization behavior. To achieve this, we could utilize other amphoteric surfactants and vary the pH and/or use a mixed system with other cationic, anionic, and nonionic surfactants that are insensitive to pH.

## Figures and Tables

**Figure 1 materials-16-03550-f001:**
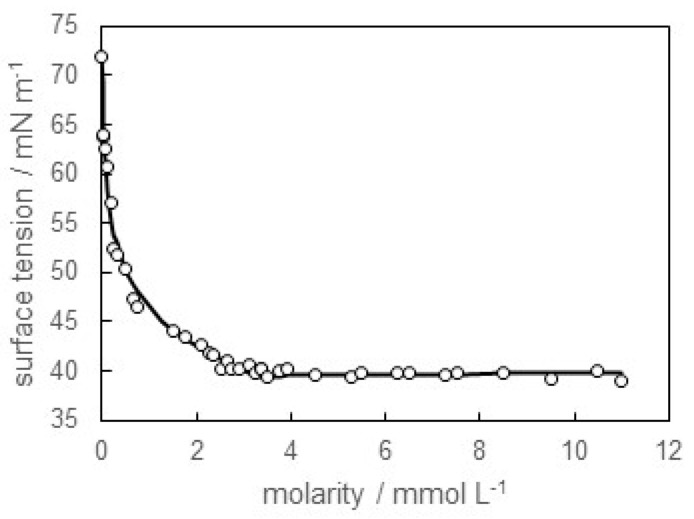
Surface tension versus molarity of SB-12 in distilled water (pH = 6.0) curve.

**Figure 2 materials-16-03550-f002:**
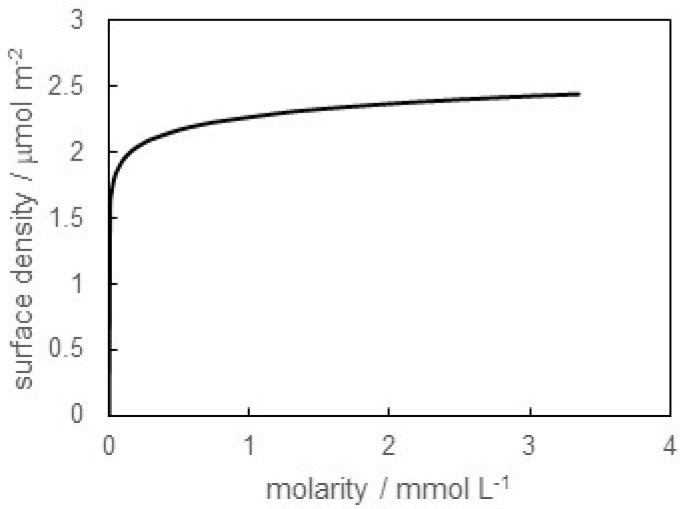
Surface density versus molarity of SB-12 in distilled water (pH = 6.0) curve.

**Figure 3 materials-16-03550-f003:**
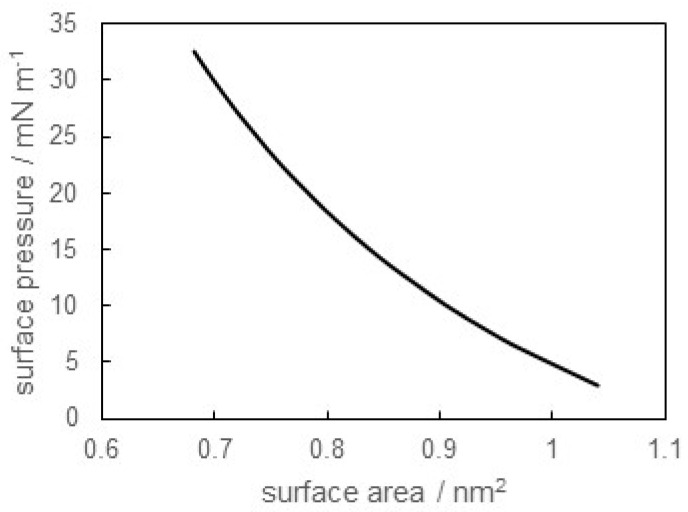
Surface pressure versus surface area occupied by an adsorbed SB-12 molecule at air/distilled water (pH = 6.0) surface.

**Figure 4 materials-16-03550-f004:**
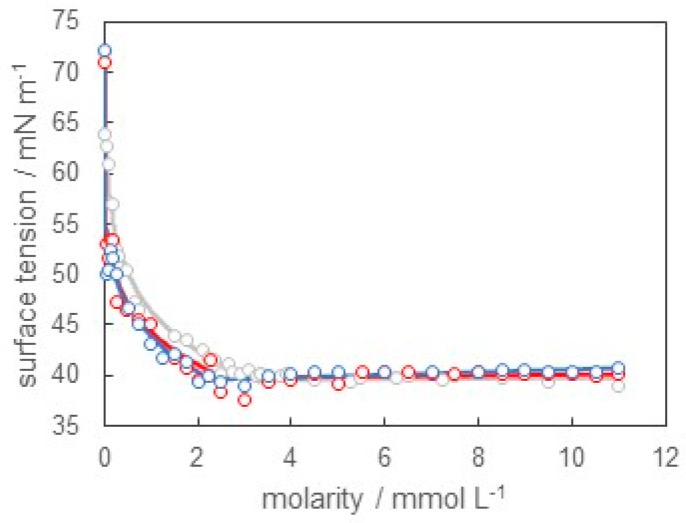
Surface tension versus molarity of SB-12 curves at pH = 1.2 (red), 6.0 (gray), and 12.5 (blue).

**Figure 5 materials-16-03550-f005:**
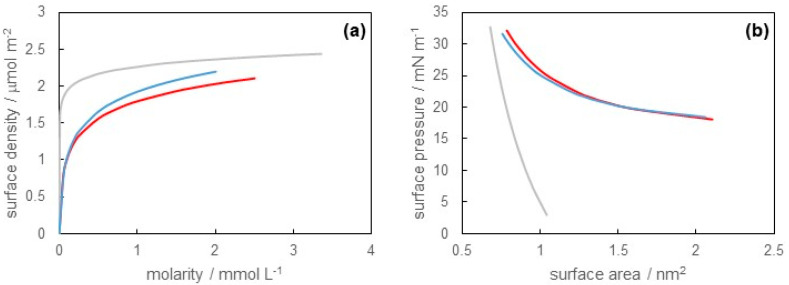
(**a**) Surface density versus molarity of SB-12 curves and (**b**) surface pressure versus surface area occupied by adsorbed SB-12 molecule curves at pH = 1.2 (red), 6.0 (gray), and 12.5 (blue).

**Figure 6 materials-16-03550-f006:**
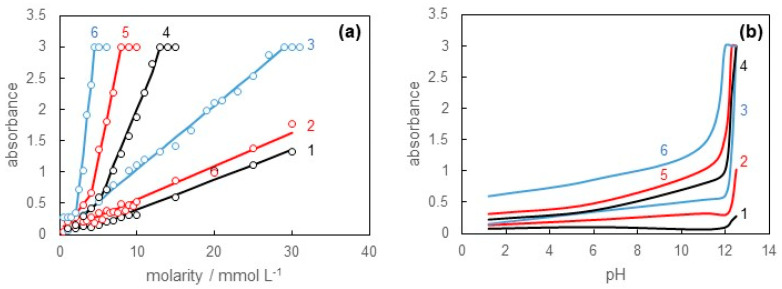
(**a**) Absorbance versus molarity of SB-12 solution curves at pH = (1) 1.2, (2) 6.0, (3) 11.0, (4) 12.0, (5) 12.3, and (6) 12.5. (**b**) Absorbance versus pH of SB-12 solution curves at constant molarity (1) 1.00, (2) 3.00, (3) 5.00, (4) 7.00, (5) 9.00, and (6) 15.0 mmol L^−1^.

**Figure 7 materials-16-03550-f007:**
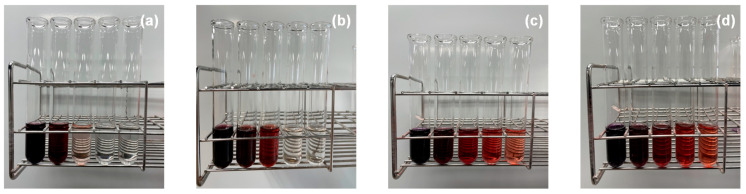
Photograph of filtrate after (**a**) 0.1 mol L^−1^ hydrochloric acid, (**b**) 0.05 mol L^−1^ hydrochloric acid, (**c**) 0.01 mol L^−1^ hydrochloric acid, and (**d**) water were added into the basic solution containing solubilized Sudan III. The volume ratio of added liquid to the whole solution, from left to right, is 0.167, 0.333, 0.500, 0.667, and 0.833.

**Figure 8 materials-16-03550-f008:**
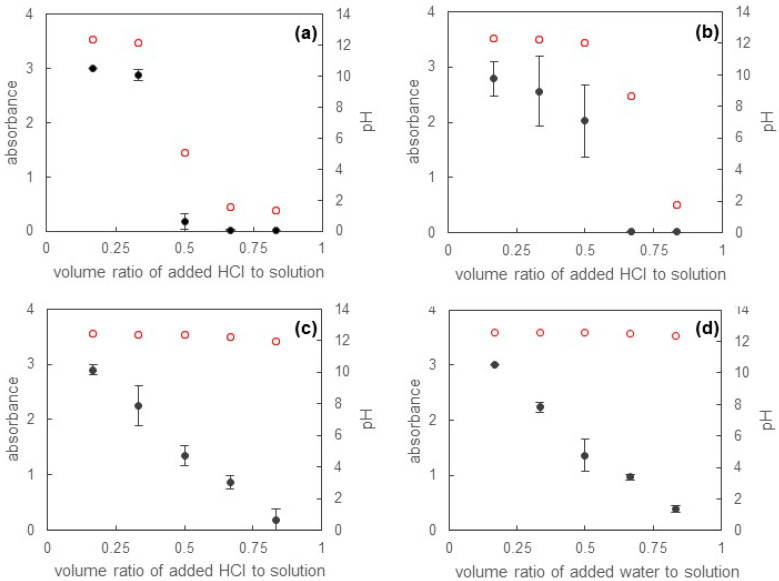
Absorbance of solution in Figure 7 after (**a**) 0.1 mol L^−1^ hydrochloric acid, (**b**) 0.05 mol L^−1^ hydrochloric acid, (**c**) 0.01 mol L^−1^ hydrochloric acid, and (**d**) water were added to the basic solution containing solubilized Sudan III. The black and red plots yield the absorbance of solution and pH values, respectively.

## Data Availability

Not applicable.

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
