# Peer review of "Study on Reversible Solubilization by Adjusting Surfactant Properties"

_materials, 2023, doi:10.3390/ma16093550_

Round 1

Reviewer 1 Report

Takata and Uchikura described "Investigation of reversible solubility by adjusting surfactant properties". The article is well written and on a relevant topic. However, while reading, a few questions and comments arose:

1) What is the reason that at pH = 6 (in Fig. 6) the dependence of surface pressure on surface area is so different from the results obtained at pH = 1.2 and 12.5? At the same time, the dependences of surface density on molarity at pH = 6, although higher, have a similar form at pH = 1.2 and 12.5 (Fig. 5).

2) It may be worth combining the numbers 5 and 6 into one, also 7 and 8. This will facilitate the perception of the material when reading.

3) Will Sudan III be removed from solution using more than 0.01 mol l-1 hydrochloric acid than shown in Figure 9? If so, at what ratio will this happen?

4) In this work there is no conclusion on the work performed.

In order to publish this work, I believe that the authors first of all need to highlight the main results of their work in the form of conclusions.

Reviewer 2 Report

This aim of this study was to control the solubilization phenomenon, especially the release of solubilized materials, by using the external stimulation.

I recommend accepting the article after performing the followig:

1- Moderate English language should be performed.

2- How can the authors explain the fact that the adsorption was not affected by the pH?

3- Please discuss why a pH less than 7 is needed for the release of SUDAN III. 

4- A conclusion section should be added to the article.

The English language needs thorough revision.

Round 2

Reviewer 1 Report

The authors did a good job, answered all the necessary questions and made changes to the article.